Living cockroach genus Anaplecta discovered in Chiapas amber (Blattaria: Ectobiidae: Anaplecta vega sp.n.)

Barna Peter 1
Šmídová Lucia smidovaluc@natur.cuni.cz 2
Coutiño José Marco Antonio 3
1 Slovak Academy of Sciences, Earth Science Institute , Bratislava , Slovakia
2 Institute of Geology and Paleontology, Charles University , Prague , Czech Republic
3 Secretaria de Medio Ambiente e Historia Natural, Museo de Paleontología Eliseo Palacios Aguilera , Tuxtla Gutiérrez Chiapas , México
Zhang Jia-Yong
Electronic publication date: 2019 Oct 28
Publication date: 2019
Volume: 7
Electronic Location ID: e7922
Received 2018 Nov 21; Accepted 2019 Sep 18
Copyright: ©2019 Barna et al.
Copyright year: 2019
Copyright holder: Barna et al.
License: This is an open access article distributed under the terms of the Creative Commons Attribution License, which permits unrestricted use, distribution, reproduction and adaptation in any medium and for any purpose provided that it is properly attributed. For attribution, the original author(s), title, publication source (PeerJ) and either DOI or URL of the article must be cited.
License URL: https://creativecommons.org/licenses/by/4.0/

Keywords: Fossil insect, Blattaria, New species, Simojovel, Cenozoic, Miocene

Funding: Slovak Research and Development Agency APVV-0436-12 UNESCO-Amba (MVTS) VEGA No. 0012-14 2/0042/18 This work was supported by the Slovak Research and Development Agency under the contract No. APVV-0436-12 and by UNESCO-Amba (MVTS); VEGA No. 0012-14, 2/0042/18. The funders had no role in study design, data collection and analysis, decision to publish, or preparation of the manuscript.

==============================
Cenozoic cockroaches are recent and with two indigenous exceptions, based on their fragmentary preservation state, they cannot be discriminated formally from representatives of living genera. Anaplecta vega sp.n. –the second described cockroach from Miocene (23 Ma) Simojovel amber (Mexico: Chiapas: Los Pocitos) is characterized by a slender, under 5 mm long body, prolonged mouthparts bearing long maxillary palps with a distinct flattened triangular terminal palpomere, large eyes and long slender legs with distinctly long tibial spines. Some leg and palpal segments differ in dimensions on the left and right sides of the body, indicating (sum of length of left maxillary palpomeres 65% longer than right; right cercus 13% longer than left cercus) dextro-sinistral asymmetry. The asymmetrically monstrous left palp is unique and has no equivalent. In concordance with most Cenozoic species, the present cockroach does not show any significantly primitive characters such as a transverse pronotum characteristic for stem Ectobiidae. The genus is cosmopolitan and 10 species live also in Mexico, including Chiapas, today. Except for indigenous taxa and those characteristic for America, this is the first Cenozoic American cockroach taxon representing a living cosmopolitan genus, in contrast with representaties of Supella Shelford, 1911 from the same amber source that are now extinct in the Americas.

Introduction

The stem Dictyoptera, containing all cockroaches, originated in the Late Carboniferous (Zhang, Schneider, & Hong , 2012; Calisto & Piñeiro, 2019) and adapted during their evolution to various environments, leading to diverse morphological adaptations including diversification of the Order Mantodea (mantises) during the Late Jurassic/Early Cretaceous (Legendre et al., 2015; Vršanský & Aristov, 2014).

Anaplecta is a cockroach genus currently placed within Ectobiidae, a family that evolved during the Mesozoic (Vršanský, 1997). Species in the subfamily Anaplectinae are small, beetle-like and mostly active during the night (Rentz, 2014). Both sexes are fully winged and can be found in the leaf litter of rainforests (Foottit & Adler, 2009).

There are several works concerning cockroaches and termites preserved in Mesozoic ambers written by Grimaldi & Ross (2004), Vršanský (2004), Vršanský et al. (2018a), Vršanský et al. (2018b), Vršanský (2009), Vršanský (2010), Anisyutkin & Gorochov (2008), Poinar Jr (2009), Vršanský et al. (2011), Vršanský et al. (2013a), Vršanský et al. (2013b), Vršanský et al. (2014), Vršanský et al. (2018a), Vršanský et al. (2018b), Vršanský et al. (2018c), Vršanský et al. (2019a), Vršanský et al. (2019b), Vršanský & Bechly (2015), Bai et al. (2016), Bai et al. (2018), Poinar Jr & Brown (2017), Sendi & Azar (2017), Šmídová & Lei (2017), Vršanský & Wang (2017), Kočárek (2018a), Kočárek (2018b), Li & Huang (2018), Mlynský, Wu, & Koubová (2019) and Podstrelená & Sendi (2018), Sidorchuk & Khaustov (2018), Qiu, Wang, & Che (2019a) and Qiu, Wang, & Che (2019b). In total, we know 11 families recorded in Mesozoic ambers out of which 3 are still living. There are 59 Cenozoic cockroaches (including the newly described A. vega) as listed in Table 1.

Table 1 List of Cenozoic cockroaches with respective literature.

Species known from sediment	Species known from amber/copal	
Blaberites rhenanaStatz, 1939	Blatta baltica Germar et Berendt 1856	
Blatta colorataHeer, 1864	B. berendtiGiebel, 1856	
B. hyperboreaHeer, 1870	B. didymaGermar & Berendt, 1856	
B. pauperataHeyden, 1862	B. ellipticaGiebel, 1862	
B.sundgaviensisFoster, 1891	B. gedanensisGermar & Berendt, 1856	
Blattidium fragileHeer, 1868	B. ruficepsGiebel, 1862	
Cariblattoides labandeiraeVršanský et al., 2011, Vršanský et al., 2012b	B. succineaGermar, 1813	
Chopardia spinipesPiton, 1940	Erucoblatta semicaecaGorokhov & Anisyutkin, 2007	
Diploptera VladimirVršanský et al., 2016	Holocompsa nigraGorokhov & Anisyutkin, 2007	
D. gemini Barna 2016	H. abbreviataGorokhov & Anisyutkin, 2007	
D. savbaVršanský et al., 2016	Latiblatta orientalisHong, 2002	
Ectobia arverniensisPiton, 1940	L. spinosaHong, 2002	
E. menatensisPiton, 1940	Nyctibora elongateStatz, 1939	
Ectobius glabellusStatz, 1939	Paraeuthyrrhapha groehniAnisyutkin, 2008	
E. kohlsiVršanský et al., 2014	Polyzosteria parvulaGermar & Berendt, 1856	
Elisama pyrulaZhang, Liu, & Shangguan , 1989	P. tricuspidataBerendt, 1836	
“Gyna” obesaPiton, 1940	Supella (Nemosupella) miocenicaVršanský et al., 2011	
Heterogamia antiquaHeer, 1849	Stegoblatta irmgardgroehniAnisyutkin & Gröhn, 2012	
Homoeogamia ventriosaScudder, 1876		
Isoplates longipennisHaupt, 1956		
Latiblattella avitaGreenwalt & Vidlička, 2015		
Morphna paleo Vršanský, Vidlička, Barna, Bugdaeva et Markevich 2013		
Paralatindia saussureiScudder, 1890		
Parallelophora acutaHaupt, 1956		
P. anomalaHaupt, 1956		
Periplaneta eocaenicaMeunier, 1921		
P. houlbertiPiton, 1940		
P. hylecoetaZhang, Liu, & Shangguan , 1989		
P. laceraZhang, Liu, & Shangguan , 1989		
P. relictaMeunier, 1921		
P. sphodraZhang, Sun, & Zhang , 1994		
Phantocephalus meridionalisZhang, Liu, & Shangguan, 1989		
Prochaeradodis enigmaticusPiton, 1940 according Cui, Evangelista, & Béthoux, 2018		
Protectobia primordialisPiton, 1940		
Protostylopyga giganteaPiton, 1940		
Pycnoscelus gardneriCockerell, 1920		
Telmablatta imparHaupt, 1956		
Zetobora brunneriScudder, 1890		
Zeunera madeleinaePiton, 1936		
Z. superbaPiton, 1940		
Pesudophylodorminae indet.Vršanský & Labandeira, 2019		

The Miocene Mexican amber originated from resinous exudates of Hymenaea sp., a leguminose tree developed near the ancient coast, in estuarine environments, very similar to mangroves (Poinar Jr, 1992) and is well studied with precise dating at 23Ma (Vega et al., 2009) and with more than 110 currently cataloged insect species (EDNA fossil insect database active 20∕11∕2018 and Vršanský et al., 2011). It was shown that the Mexican amber assemblage is similar to the modern assemblages recorded from the sticky traps (Solórzano Kraemer et al., 2018). Cockroaches are represented by the genus Ischnoptera Burmeister, 1938 reported by Solórzano-Kraemer (2007), although the identification needs further support) and Supella miocenica (Vršanský et al., 2011).

The extant genus Anaplecta is today a widely distributed circumtropic taxon (see Beccaloni, 2014) and its ecology remains very little known. Fossils of the genus Anaplecta, aside from Mexican amber are also known from Eocene Baltic amber and Chinese ambers (P. Barna; 1.11.–20.11.2013, Moscow, PIN RAS and private collection V. Gusakov, specimen number PIN 964/1322; personal observation) and undescribed Anaplecta are also reported from Dominican amber (Gutiérrez & Pérez-Gelabert, 2000), but it is unclear whether the mentioned specimens do not represent the common Plectoptera electrina Gorokhov et Anisyutkin in Gorokhov (2007)— locations are marked in Fig. 1E.

Figure 1 Photos of amber inclusion and map of destribution of genus Anaplecta spp.

(A) Partial 3D extraction. (B) Ventral view. (C) Dorsal view. (D) Whole piece of amber, ventral view. Specimen overall length head-abdomen, 4.89 mm. (E) Distribution map of amber Anaplecta spp. with the Baltic amber reaching out of the present range.

Material and Methods

The studied holotype of Anaplecta vega, sp.n. (catalogue number IHNFG-5323) comes from Miocene (23 Ma) Simojovel amber (Mexico: Chiapas), Los Pocitos (92°43″46″W, 17°08″53″N). The specimen is stored in Paleontological Collection of the Museo de Paleontología “Eliseo Palacios Aguilera”, ascribed to Secretaría de Medio Ambiente e Historia Natural (SEMAHN) of the Chiapas Government, Mexico under the catalogue number IHNFG-5323.

The specimens Anaplecta xanthopeltis Hebard, 1921 (MNHN-EP-EP1398, 2018) and Anaplecta maronensis Hebard, 1921 (MNHN-EP-EP1385, 2018), used for comparison with living Anaplecta, are deposited at the National Museum of Natural History in Paris, France.

Photographs were taken with a KEYENCE digital microscope, which took pictures from different locations and focal depth and then automatically combined them into a single stacked photo. This type of picture was also used as a basis for making a highly detailed line drawing in CorelDrawX3, where we used additional photographs of separate parts of the cockroach body. These were taken with a LEICA MZ6 binocular loupe and LEICA EC3 camera. The dorsal drawing was manually made using a drawing ink pen applied over a transparent paper.

Abbreviations used: l, length; w, width (all in mm).

The electronic version of this article in Portable Document Format (PDF) will represent a published work according to the International Commission on Zoological Nomenclature (ICZN), and hence the new names contained in the electronic version are effectively published under that Code from the electronic edition alone. This published work and the nomenclatural acts it contains have been registered in ZooBank, the online registration system for the ICZN. The ZooBank LSIDs (Life Science Identifiers) can be resolved and the associated information viewed through any standard web browser by appending the LSID to the prefix http://zoobank.org/. The LSID for this publication is: [article: urn:lsid:zoobank.org:pub:FD6F76DB-BF88-4FBA-8737-F00B408C54E1]. The online version of this work is archived and available from the following digital repositories: PeerJ, PubMed Central and CLOCKSS.

Figure 2 Line drawing of Anaplecta vega sp.n.

Systematic Palaeontology

Order Blattaria Latreille, 1810 (= Blattodea Brunner von Wattenwyl, 1882)	
Family Ectobiidae Brunner von Wattenwyl, 1865	
Subfamily Anaplectinae Walker, 1868	
Genus Anaplecta Burmeister, 1838	

Type species: Anaplecta lateralis Burmeister, 1838

Composition. An up-to date list can be found in the online database ‘Cockroach Species File Online’, which was founded by Beccaloni (2014) based on the world catalogue of cockroaches compiled by Princis (1962; 1963; 1964; 1965; 1966; 1967; 1969; 1971).

Occurrence. Circumtropical; during Eocene also in Baltic, Dominican and China areas (in preparation by authors), which had a subtropical climate that time. Stratigraphic range: Eocene-living.

Anaplecta vegasp.n.	
(Figs. 1A–1D, 2A–2C)	

Types. One complete adult specimen (Holotype kept in Paleontological Museum in Tuxtla, Mexico) with folded wings, probably male, enclosed in a small piece of amber. Catalogue number IHNFG-5323.

Type horizon and locality. Lower Miocene, Mazantic Shale. Los Pocitos locality NW from Simojovel de Allende in Chiapas, Mexico. 92°43″46″W, 17°08″53″N.

Material. Type only.

Etymology. After VEGA (Vedecká Grantová Agentúra –Research grant agency of the Slovak Republic) and also after Dr. Francisco Vega (UNAM, Mexico city) who did so much for progressing research on Chiapas amber.

Differential diagnosis. Small slender roach with body l = 4.89 (excluding antennae and cerci) and w = 2.00; subtriangular rounded pronotum; prolonged head with unique large eyes and huge asymmetrical maxillary palps; antennae similar in length as the body; tegmina reaching apex of abdomen; long slender legs bearing long tibial spines.

Differs from all species, except for A. xanthopeltis, in having a derived simplified form of the pronotum (without paranotalia and transverse shape) and, except for A. maronensis, a derived reticulated forewing venation.

Differs from recent species from Mexico (since this genus contains a large number of species worldwide and no other fossil species of this genus were described from this area): A. azteca (Saussure, 1868) is larger, its pronotum is double the length and width of the pronotum of A. vega, the pronotum length is 1/3 of tegmina length, while in A. vega it is 1/4.

A. fallax Saussure, 1862 has quite a similar tegmina length and total body length, but the pronotum is distinctly larger, can reach almost two times the dimension of the pronotum of A. vega; the pronotum length can be more than 1/3 or even 1/2 of the tegmina length. Anterior margin ascending under sharper angle from the distal third of tegmen length forming a rounded angle, while in A. vega it is at the beginning of the most distal fifth of the tegmen length.

A. mexicana Saussure, 1868; while having a similar tegmina length and pronotum length ratio, the whole body is significantly larger and the tegmina length is double the tegmina length of A. vega. The shape of tegmina differs with the anterior margin slightly sinusoid without any pronounced angulation, the tegmina apex is more broadly rounded and positioned around the middle of the tegmen width. A. nahua Saussure, 1868 is larger, the tegmina length: pronotum length ratio is quite similar to that of A. vega.

A. otomia Saussure, 1869 is larger, dark colored, pronotum with nearly opaque lateral margins, tegmina in apical third strongly narrowed unlike A. vega. Its anterior margin does not look angular in the apical fifth and is curved smoothly, and the radius area of the tegmina is much narrower.

A. saussurei Hebard, 1921 has a similar size and similar shaped tegmina as A. vega, but the tegmina reach slightly beyond the cercal apices, are slightly wider, their anterior margin in the basal part is almost straight, the clavus is distinctly longer and wider, the pronotum is larger and the pronotum length is 1/3 of the tegmina length.

A. tolteca Saussure, 1868 is larger but the tegmina length: pronotum length ratio is the same as in A. vega.

A comparison of dimensions of A. vega n. sp. and the above mentioned Mexican species is presented in Table 2A; a comparison of A. vega n. sp. dimensions and an average of the mentioned Mexican species dimensions is shown in Table 2C.

Table 2 Measurements of Anaplecta vega.

(A) Body measurements of Anaplecta vega sp.n. and different Anaplecta species. (B) Comparison of Anaplecta vega sp.n. left and right maxillary palpomere lengths. (C) Comparison of A. vega sp.n. dimensions and average of living Mexican Anaplecta species dimensions. (D) Comparison of length of each leg femur, tibia and tarsomeres of Anaplecta vega sp.n. (E) Left and right antennomeres length comparison including scape and pedicel present in Anaplecta vega sp.n.

	Body l	Tegmina l	Pronotum l	Pronotum w	Tegmina l/pronotum l	Pronotum l/pronotum w	
A	
A. vega including tegmina	4,89	4,04	1	1,27	4,04	0,79	
A. azteca	6,5	5,8	2	2,5	2,9	0,8	
A. fallax	4,6	4	1,5	2,25	2,67	0,67	
A. decipiens male incl. tegmina	4,8	3,5	2	2,4	1,75	0,83	
A. decipiens female incl. tegmina	5,8	4,6	2	2,4	2,3	0,83	
A. mexicana	8	8,5	2	3,5	4,25	0,57	
A. gemma	6,7	6,8	1,7	2,3	4	0,74	
A. nahus incl. tegmina	6	5,5	1,5	2,4	3,67	0,63	
A. otomia	6,5	6	1,75	2,4	3,43	0,73	
A. saussurei	4,1	3,8	1,3	1,9	2,92	0,68	
A. tolteca	6	6,5	1,6	2,5	4,06	0,64	
	Right maxillary palp	Left maxillary palp	
B	
1	0,13	0,13	
2	0,1	0,14	
3	0,31	0,4	
4	0,17	0,29	
5	0,27	0,34	
	A. vega length (including tegmina)	Avarage Mexican Anaplecta length	
C	
Body l	4,89	5,9	
Tegmina l	4,04	5,5	
Pronotum l	1	1,74	
Pronotum w	1,27	2,48	
Tegmina l/pronotum l	4,04	3,19	
Pronotum l/pronotum w	0,79	0,71	
	Front L	Front R	Middle L	Middle R	Hind L	Hind R	
D	
Femur	1,29	0,67	1,29	0,91	1,38	1,4	
Tibia	0,89	0,73	0,89	1,11	1,89	2,13	
Tarsomere 1	0,34	0,33	0,34	0,56	0,91	0,84	
Tarsomere 2	0,07	0,08	0,07	0,12	0,19	0,19	
Tarsomere 3	0,06	0,06	0,06	0,08	0,11	0,11	
Tarsomere 4	0,07	0,06	0,07	0,04	0,1	0,1	
Tarsomere 5	0,11	0,2	0,11	0,18	0,18	0,18	
	Left antenna	Right antenna	
E	
1	0,33	0,29	
2	0,22	0,2	
3	0,16	0,16	
4	0,09	0,09	
5	0,09	0,07	
6	0,09	0,07	
7	0,09	0,1	
8	0,1	0,1	
9	0,11	0,12	
10	0,12	0,11	
11	0,12	0,12	
12	0,13	0,14	
13	0,14	0,13	
14	0,14	0,14	
15	0,14	0,13	
16	0,14	0,14	
17	0,16	0,16	
18	0,16	0,16	
19	0,16	0,16	
20	0,16	0,14	
21	0,14	0,17	
22	0,17	0,17	
23	0,18	0,17	
24	0,17	0,17	
25	0,16	0,13	
26	0,17	0,16	
27	0,16	0,18	
28	0,16	0,19	
29	0,16	0,17	
30	0,16	0,16	
31	0,16	?	
32	0,15	?	
33	0,14	?	
34	0,1	?	

Description. Detailed measurements are in the Table 2.

Body small and slender (l = 4.89, w = 2.00), tegmina reaching apex of abdomen, legs long and slender carrying large tibial spines, antennae similar in length to the body.

Pronotum subtriangular, rounded, cranially arched over head (l = 1.00, w = 1.27), long erect setae sparsely distributed along pronotum margin and on its dorsal surface. Pronotum length is 1/4 of tegmina length.

Scutellum triangular, cranio-caudally prolonged (length of scutellum part not covered by pronotum = 0.29, w = 0.14).

Tegmina total l = 4.04, l of part not covered by pronotum = 3.89, left tegmen w = 1.28; visible left clavus l = 1.16, w = 0.67. Basal half of tegmina inflated with exception of anterior peripheral areas (costal area, part of radial area). Anterior margin in the apical fifth of tegmen length starts to tilt posteriad more strongly, which gives it an angular look. Apex postero-apically sharpened. Costal area wide. Radial field in apical half wide, branches of radius almost all simple (one secondary dichotomy observed in right tegmen, left tegmen veins weakly visible). Surface sclerotized, but not fully elytrized, without prominent structures. Sparsely distributed medium sized setae occur at anterior and apical margins of tegmina and medium sized to long setae on tegminal veins. Only a very small portion of right tegmen is covered by left tegmen. Clavus l = 1.16 (only the visible uncovered part), w = 1.23.

Hind wings covered by tegmina, folded in half as standard for the genus.

Head with prolonged mouthparts and large eyes, which in lateral view cover almost the entire head excluding mouthparts; head w = 0.76, length from top of vertex to distal part of mandibles = 0.91, distance from occipital foramen to top of frons = 0.47; eyes subovoid in lateral view, eye length (parallel to head length) = 0.42, eye width (perpendicular to eye l) = 0.34, interocular w = 0.4 mm. Three medium-sized setae positioned between left eye ommatidia near gena . Vertex sparsely covered by setae ranging from short to very long; frons and clypeus with few distinctly long setae, gena posteriorly with three distinct medium sized setae and smaller thin setae along eye margin. Maxillary palps long with broad triangular terminal segment; dimensions of palpomeres of right and left maxillary palp differ (right 1st palpomere l = 0.13, w = 0,06; left 1st palpomere l = 0.13, w = 0,09; right 2nd palpomere l = 0.1, w = 0.05; left 2nd palpomere l = 0.14, w = 0.07; right 3rd palpomere l = 0.31, w = 0.06; left 3rd palpomere l = 0.4, w = 0.09; right 4th palpomere l = 0.17, w = 0.07; left 4th palpomere l = 0.29, w = 0.14; right 5th palpomere l = 0.27, w unmeasurable due to position of the cockroach body in the amber; left 5th palpomere l = 0.34, w = 0.21, apical contacting surface 0.34; plot of left and right palpomeres length comparison is shown in Table 2B. Labial palps considerably smaller than maxillary palps, terminal palpomere triangular, distally widened. Only 2nd and 3rd left labial palpomere sufficiently visible to be measured: 2nd left palpomere l = 0.11, 3rd left labial palpomere l = 1.13.

Antennae length similar to body length. First three antennomeres only with few setae, more distal antennomeres are richly covered by distinct setae, which exceed and in some parts double the width of antennomeres.

Scape large (left l = 0.33; right l = 0.29) with wide proximal half (left w = 0.13) and sharp transition into narrower distal half (w = 0.09), distal ending oblique with five setae.

Pedicel cylindrical (left l = 0.21, left w = 0.08; right l = 0.2), distal end oblique with distinct sharp angle at one side and wider than proximal end. Proximal third swollen on one side. Setae very few in number.

Segments of flagellum, 32 in left antenna, 28 in right antenna. Each segment is more or less wider in its distal part than in its proximal part. Setae longer than the width of flagellar segments.

First flagellar segment (third antenomere) slightly elongated, length is almost two times as its largest width (left l = 0.14, w = 0.07). Following basal flagellomeres are short almost square shaped flagellomeres only slightly longer than wide. The subsequent row of flagellomeres has a lengthening trend distad.

Table 2E shows a chart that compares left and right antennomeres lengths of individuals.

Cerci 7-segmented. 1st cercomere and 7th cercomere thinner than the rest, while the 7th distinctly tapers distally forming a sharp end, cercomeres 2-6 sublenticullar, being similar in shape and size. Left cercus total l = 1.00, right cercus total l = 1.13; Setae on cercomeres have different dimensions, from small ones ranging from 1/3 or 1/4 of the cercomere length to thicker prominent setae approximately the size of the cercomere length (maximal l = 0.21, but majority around 0.14) and long thin setae the size of two or three cercomeres which occur at the rate of 1 or 2 per cercomere. The entire surface of the cerci is also covered by very small short microsetae.

Legs slender and very long (hind legs longer than body) with large spines (longer than tarsomeres, except the 1st tarsomere) on tibia and distal end of femur.

Fore coxae subtrigonal with convex anterior margin, widest before middle , more slender in distal half. Few setae present along the posterior margin.

Fore trochanteri very thin, barely visible.

Fore femora slender (left fore femur l = 1. 29, w = 0.2, right fore femur l = 0.67, maybe more, visibility obscured by damage of amber) with subparallel ventral and dorsal margins, which are only slightly convex, narrower in proximal part and in distal third of their length. Anteroventral margin in distal half with 20 ± shorter spines, posteroventral margin with 13 (observed) longer setae sparsely distributed along fore femur length, two thicker spines present on anterior surface of proximal half of fore femur. Terminally three long serrated spines present: anteroventral, posteroventral, anterodorsal. Rest of forefemoral surface covered only by a low number of shorter setae, mostly concentrated in dorsal part.

Fore tibiae distinctly more slender than fore femora, generally retaining similar width (w = 0.07) throughout its length (left fore tibia l = 0.89, right fore tibia l = 0.73), except for the thinner arched proximalmost spineless part (w = 0.06), and neglectable changes of width due to elevations around large, articulated, serrated spines; these spines are up to 0.29 long and 0.01 wide with three spines in the middle third of the tibial length and two peripheral spines facing dorsad, the middle one facing posteriad. Four large spines are at the distal end of the tibia (1 anteroventral, 1 anterior, 1 dorsal, 1 posterior). Distribution of large spines is the same on both right and left fore tibiae. Along dorsal and ventral side and distal half of anterior surface are sparsely distributed medium-sized setae.

Fore tarsi being 5-segmented, very slender, covered by setae exceeding their width (w = 0.04), terminated by trilobal arolium and two thin arcuate, more or less symmetrical claws with widened bases.

Middle coxae (distal part obscured by damage in amber and another leg) larger and wider than fore coxae, on posterior margin few setae present; distal end has distinct smaller lobe with 6 longer setae.

Middle trochanteri wide (width is only a little less than length), slightly curved.

Middle femora elongated with slightly convex dorsal and ventral side (ventral side being almost straight) with bigger width around the middle of length (left middle femur l = 1.44, w = 0.22 mm; right middle femur l = 0.91, w = 0.17). Setae sparsely distributed around dorsal margin, ventral margin with 6 larger thick setae and 1–2 medium-sized setae between each two consequent larger setae; on proximal half of middle femur present anteroventrally two longer, anterad facing spines; on distal end of middle femur present 3 large serrated spines, one anteroventrally, one posteroventrally and one dorsally.

Middle tibiae similar in length (left tibia l = 1.16, right tibia = 1.11), but left tibia 1/6 shorter than left middle femur, right tibia 1/5 longer than right middle femur; width varying along tibial length as result of elevations at the bases of spines (minimum w = 0.07, maximum w = 0.1); 10 large serrated spines (maximal l = 0.36, w = 0.01) facing differently (on left middle tibia 1 anteroventral, 1 posteroventral, 4 anterodorsal 4 posterodorsal) along tibial length, 4 terminal large serrated spines (anteroventral, posteroventral, posterodorsal, anterodorsal) and one shorter spine (posteroventral).

Middle tarsi 5-segmented, slender, covered by medium sized setae exceeding tarsal width (w = 0.04 mm), terminated by two arched slender claws with widened bases and trilobal arolium.

Hind coxae badly visible.

Hind trochanteri slender, slightly curved with few setae, left hind trochanter l = 0.43, w = 0.11; right hind trochanter l = 0.46, w = 0.09.

Hind femora larger than fore and mid femur (l = 1.4, w of right hind femur = 0.31, left one in wrong position to be measured) with biggest width in the middle of their length, distal end slightly widened, ventral side only slightly convex, dorsal side more convex. Numerous short setae scattered through whole surface of femur. Setae sized from short to long present along dorsal femoral margin, getting longer distad; at anterior surface setae with dark bases present at an arched line subparallel to the dorsal side of femur, which proximally starts around the middle of hind femur width, approaching dorsal side of hind femur distad. Dorsal and anterior setae are longer on left fore femur. Anteroventral edge with medium-sized setae and two large spines in the middle third of hind femur length. Posteroventral edge with five long setae (left hind femur) and shorter setae between them. Terminally three long serrated spines present: anteroventral, posteroventral, anterodorsal.

Hind tibiae longer than fore and mid tibiae (left hind tibia l = 1.89, right hind tibia l = 2.13) near double length of middle tibiae, width of hind tibiae is weakly varying due to elevations at bases of larger spines, but not showing a significant narrowing or widening trend (maximal w = 0.13), exception is the proximal 1/6 which is more slender (w = 0.1 mm). Each hind tibia has along its length 17 long serrated spines (three anteroventral, two posteroventral, seven anterodorsal, five posterodorsal) and five long serrated terminal spines (1 anteroventral, two posteroventral, one posterodorsal, 1 anterodorsal), which makes together 22 spines on one hind tibia (maximal spine l = 0.5 mm maximal w = 0.02). Medium sized seatae are sparsely distributed along ventral and dorsal margin (up to 4 between two spines).

Hind tarsi larger than fore and mid tarsi, 5-segmented, slender, covered by medium sized setae most of which equal in size or exceeding tarsal width (w = 0.06), terminated by two arched slender claws with widened bases and trilobal arolium (pulvilli absent or indistinct). 1st tarsomere very longwith almost same width throughout its length, in proximal part slightly thinner; covered by distinct medium sized setae, most prominent is the ventral row of setae, other areas have less densely distributed thinner setae; subsequent tarsomeres have the same length on left and right hind leg.

Comparison of length of each leg femora, tibiae and tarsomeres is visible in Table 2D.

Occurrence. Lower Miocene, 23 Ma. Chiapas amber, Mexico.

Discussion

The assignment of the herein described specimen to the subfamily Anaplectinae is in accordance with Grandcolas (1996). The majority of autapomorphies listed there are observable also in A. vega. It is in general small-sized species, other small-sized dictyopterans were observed and discribed in Vršanský (2002). Long, straight setae on pronotum, strong sharp apophysis on the anterior arch and transverse folding of hind wings are present. However, the vannus with short successive dichotomies on its first vein was not observed. The studied specimen was assigned to the genus Anaplecta on the basis of the overall body shape, shape of the pronotum, smooth dorsal half of the body, large axe-like terminal mandibular palpomere (however, it does not have the same length as the former palpomere, as mentioned in the original description of the genus), coriaceous tegmina, hind wings folded in half, arrangement of femoral setae and spines, large cerci, and tarsomeres without pulvilli. The specimen is most similar to and compared with the holotypes of South American A. xanthopeltis Hebard, 1921 and A. maronensis Hebard, 1921, which have the same type of pronotum, overall tegmina shape, tegmina type of venation and hind wings folding. Anaplectinae is usually considered a problematic group with an interesting systematic background. It is manifested both in morphological and molecular analysis as a “long branch”, the most basal, earliest derived group among extant standard cockroaches according to the molecular analyses (Klass & Meier, 2006; Legendre et al., 2015). In contrast, the fossil record reveals earlier branched Ectobiidae and Blattidae (Sendi & Azar, 2017) and Anaplecta is considered more recent.

It is an interesting fact, that the evolution of this group is accelerated, which is not only reflected in sclerotisation of the fore wing, but also in simplification of venation, which probably happened due to miniaturization of the body. The large asymmetry in leg dimensions on the left and right sides of the body (Table 2D) and distinctly larger left maxillary palpomere (Table 2B) are also evidence of nonstandard evolution of the genus Anaplecta.”

The coriaceous tegmina are not sclerotized completely enough to be considered as elytrised, which is also seen in living A. maronensis.

In respective descriptions of living Mexican species the data about the ecology of Anaplecta are missing. The diversity of the genus is very high in rainforest areas. Evangelista et al. (2015) highlight four species in Amazonas (Venezuela), four species in Guyana, 10 species in Suriname, nine species in French Guiana, 10 species in Ampa (Brazil); Vidlička (2013) mentions 10 species in Ecuador; eight species (including the new species) are known from Mexico based on the works of Saussure (1862), Saussure (1868), Saussure (1869) and Hebard (1921).

The genus Anaplecta has presently a circumtropical distribution, and the (sub)tropical climate supports the newly described species as well. Fossil Anaplecta species are also known from Eocene Baltic Kaliningrad and Chinese ambers—the climate during Eocene in these areas was subtropical (Grimaldi, 1996), and from the related Dominican amber (nevertheless, the Dominican species are undescribed and taxonomical placement needs confirmation—see above).

The palaeogeographical inferences are principal, as it has been shown that the Eocene North American fauna (major locality Green River, Colorado, U.S.A. but also more northern localities in Canada—Greenwood et al., 2005, Archibald & Mathewes, 2000) and also the Miocene fauna of Chiapas amber was cosmopolitan, while younger Dominican amber contains modern, American cockroach taxa—strongly suggesting a major extinction between these two time periods (of deposition of these two sites—Vršanský et al., 2011). The present study cannot reveal information on whether Anaplecta, inhabiting Americas today, is a native trace of the original Eocene diversity or a descendant of a more recent re-invasion. This research awaits future investigation.

The detailed phylogenetic study of Djernaes, Klass, & Eggleton (2014) and Vidlička et al. (2017) positioned the Anaplectidae into one clade together with Tryonicidae, Cryptocercidae and Isoptera based on what seems to be a very primitive taxon. However, according to our morphological and taphonomical (i.e., absence in the rich Mesozoic record counting 30,000 sedimentary and over 3,000 amber specimens) observations, Anaplecta is a modern (plesiomorphy such as non-fully elytrised tegmina of the present species are also shared with some living representatives—see above) and developed genus typical of the Cenozoic.

The asymmetry of genitalia is common and appeared repeatedly during the insect evolution (Huber, Sinclair, & Schmitt , 2007), it is even part of the original ground plan in the whole order Dictyoptera. Asymmetries in left–right axis of other body parts can be also found among insects (Smith, Crespi, & Bookstein , 1997). The fluctuating asymmetry can predict developmental instability of the individual (Dongen, 2006). In that case, the studied individual was vital and the unevenness of certain body parts did not affect the fitness or if it did, not fatally. The difference between left and right hind legs is neglectable (Femur (F) = 1.014[r]/ Tibia (T) = 1.12[r]/ Tarsomere (Ta) 1 = 1.08[l]/ Ta2 = 1/Ta3 = 1/Ta4 = 1/Ta5 = 1). It explains the importance of the hind leg in the movement (Hughes, 1952). In contrast, the front femur leg (also important in the movement) is highly asymmetrical. The biggest difference can be found between the left and right femur (the left femur is almost twice as big). Also, the overall asymmetry is more evident (F = 1.93[l]/ T = 1.2[l]/Ta1 = 1.03[r]/Ta2 = 1.14[r]/ Ta3 = 1/Ta4 = 1.16[l]/Ta5 = 1.81[r]). The most asymmetrical tarsomeres can be observed in the middle leg (F = 1.41[l]/ T = 1.25[l]/ Ta1=1.65[r]/ Ta2=1.71[r]/Ta3=1.33[r]/Ta4=1.75[l]/Ta5=1.63[r]).

The expansion of extremities longitudinally could have been caused post mortem, by tension of polymerizing resin. The structure of resin can be modified due to conditions such as temperature, humidity change, etc. These changes can affect the state of preservation of the inclusion (Poinar Jr & Mastalerz, 2000).

The irregularity in length of extremities could have also happened while attempting to escape after being embedded in resin, which often even leads to disarticulation (Martínez-Delclòs, Briggs, & Peñalver , 2004).

Conclusions

The Miocene cockroach Anaplecta vega sp.n. represents an extinct species of an extant genus and is consistent with a cosmopolitan pattern of Cenozoic occurrences. Its closest relatives live in South America, with which it shares the pronotum shape, tegmina shape and venation, and hind wing folding. Since the living representatives of Anaplecta prefer warm climates, Anaplecta vega probably also lived in warm (sub)tropical areas. It is the second cockroach species described from Chiapas amber, Mexico and belongs to the subfamily Anaplectinae, family Ectobiidae. The described individual shows noticeable asymmetries in maxillary palpomeres length, right cercus and some leg segments. The asymmetry regarding the genotypic or phenotypic origin remains obscure and needs further study.

Additional Information and Declarations

Competing Interests

Author Contributions

Data Availability

New Species Registration

The authors declare there are no competing interests.

Peter Barna conceived and designed the experiments, performed the experiments, analyzed the data, contributed reagents/materials/analysis tools, prepared figures and/or tables, approved the final draft.

Lucia Šmídová performed the experiments, prepared figures and/or tables, authored or reviewed drafts of the paper, approved the final draft.

Marco Antonio Coutiño José conceived and designed the experiments, prepared figures and/or tables, approved the final draft.

The following information was supplied regarding data availability:

The following information was supplied regarding the registration of a newly described species:

Publication LSID: urn:lsid:zoobank.org:pub:FD6F76DB-BF88-4FBA-8737-F00B408C54E1

Species LSID: Anaplecta vega, sp.n. urn:lsid:zoobank.org:act:76FFB45B-41C4-4E32-A115-9037EF1EFA86

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
