# Peer review of "Living cockroach genus Anaplecta discovered in Chiapas amber (Blattaria: Ectobiidae: Anaplecta vega sp.n.)"

_PeerJ, doi:10.7717/peerj.7922_

## Round 0.1 · original submission · Major Revisions

Dear Dr. Smidova,

Thank you for your submission to PeerJ.

I appreciate that there are six reviewers' comments, however we were receiving conflicting feebback and so, in discussion with the Section Editors for the journal, we sought more feedback than would normally have been the case.

I think the comments from Reviewers 5 and 6 are pretty similar in terms of broad suggestions. Critically, I agree that the referral of the specimen to Anaplecta must be better supported -this is essential for the revision, in my view. The concerns about the quality of the literature review and breadth of cited literature are also valid, in my reading of the situation.

The reviewers‘ comments will be helpful to improve your paper.

We look forward to your revision.

With kind regards,
Jia-Yong Zhang

·

Basic reporting

I recommend to fuse figures 1+2 (1) and all others except illustrations (2, 3) in total 3 figures, only first of which needs color. But this is not obligatory.
Some references are differently formatted.

Experimental design

no comment

Validity of the findings

no comment

Additional comments

Great job, congratulations

Reviewer 2 ·

Basic reporting

The paper is properly organized, clearly and the English language well (with some very minor exceptions, see the review)

Experimental design

no comment

Validity of the findings

Authors gave the differences of the new species with all other species, and described the structure of the species in detail. It is validity to erected a new species.

Additional comments

This is an interesting paper described a new species and species based on one specimen which come from Miocene Simojovel amber. We can know more informations about Miocene through amber. Whereas I found some minor errors in your paper, please correct them.

Annotated reviews are not available for download in order to protect the identity of reviewers who chose to remain anonymous.

Reviewer 3 ·

Basic reporting

no comment

Experimental design

no comment

Validity of the findings

no comment

Additional comments

See comments in the PDF

Annotated reviews are not available for download in order to protect the identity of reviewers who chose to remain anonymous.

·

Basic reporting

engish clear
references ok
structure ok
results coherent

Experimental design

aim and scope attained
question well defined
investigation rather rigorous, some points need better justification
methods well described

Validity of the findings

new
data robust
conclusions should be better supported

Additional comments

Cenozoic cockroaches were modern and with two indigenous exceptions they represent living genera

This sentence is curious, as it is what s currently known, not all the taxa !!, also, the fossils have been attributed to extant genera but on the basis of generally incomplete fossils, especially the genitalia are nearly always missing
So be more prudent

cockroach does not show any significantly primitive characters
on the basis of which phylogenetic analysis, please precise

Order Blattodea (cockroaches) originated in Late Carboniferous
No, no proof at all,
Stem dictyoptera are known in carboniferous, which is very different
replace by stem group of Dictyoptera

line 41: reorder the references from oldest to most recent
Föster

Line 76: strange sentence, extraction of what ? rephrase please

it is unclear whether the mentioned specimens does not represent

line 141: in having derived simplified form of pronotum
precise why derived, and simplified in what ?

line 328: please give at least one reference to justify the genus attribution, and precise which are synapomorphies, and why (reference of a phylogeny), important point !!

of Chiapas amber were cosmopolitan, while younger Dominican amber contain modern
please, give reference on the dating of the these ambers, because age of Dominican amber is controversial ….
Line 359: why Eocene, you have fossils from this period ?

Reviewer 5 ·

Basic reporting

In this manuscript, the authors describe a new extinct species of cockroach, Anaplecta vega, from one specimen preserved in Chiapas amber. This study is quite straightforward but it lacks some context and references, and some figures are irrelevant or not clear enough (see general comments for more details)

Experimental design

The methods is not described with sufficient details (see general comments for more details)

Validity of the findings

The figures does not allow to observe the characters discussed in the manuscript.
A list of characters justifying the taxonomic position of the new species within the genus Anaplecta is missing.

Additional comments

In this manuscript, the authors describe a new extinct species of cockroach, Anaplecta vega, from one specimen preserved in Chiapas amber. They provide a lengthy description of the head, tegmina, legs… and a differential diagnosis with extant Mexican Anaplecta species. This work is interesting and quite straightforward but I have a few concerns preventing the publication of this manuscript.

1. The authors must provide a list of clearly identified characters to justify the taxonomic position within the genus Anaplecta. In addition, the description relies only on a single specimen, which is always problematic although acceptable. Still, this specimen has asymmetrical hindlegs and palps. The author should explain why they believe this asymmetry is characteristic of the species and not specimen-specific (e.g. due to taphonomic processes or due molting issues, etc.)

2. The introduction should be revised because its structure is not clear and its content inaccurate.
The first paragraph, composed of a single confusing sentence (Mantodea does not belong to Blattodea, they are sister-groups), misses several references (and Brongniart 1885 is inadequate given the progresses made in our understanding of roachoid fossils and their relationship to modern Dictyoptera).
The second paragraph is supposed to be a list of works concerning cockroaches. Yet, some studies concern Alienoptera, Raphidiomimidae, which are not cockroaches sensu stricto, or Aethiocarenodea, which are not cockroach at all. This list, supposed to be exhaustive(?) and in alphabetical order(?) must be revised.
The third paragraph is also a list of studies, but about Cenozoic cockroaches. Among this list, the authors could distinguish the ones relying on amber as their introduction seems to emphasize fossils from ambers.
The fourth paragraph is again a list: list of species from the studies listed in the previous paragraph. This list is not easy to read and could be presented in a Table instead.

3. Some figures are irrelevant or do not allow observing the characters discussed in the paper. Specifically, several characters (in the tegmina, the legs) cannot be seen in Figures 2 and 3. I understand that good pictures might be challenging to obtain but the authors should provide better figures (close-ups) so that the reader can observe the characters and not only rely on the authors’ description. In Figure 5, computing average measurements for living Mexican Anaplecta species does not bring much useful information. Why is it useful to know that Anaplecta vega is smaller than the average of Mexican Anaplecta species? In Figures 7 and 8, it is unclear what the authors want to show. In addition, some measurements are doubtful as noticed in the text but not in the figure; which is quite misleading (e.g. femur of right front leg).

4. In the methods, the authors mention two species of Anaplecta that have been used for comparisons. However, it is unclear whether the authors borrowed and looked at the type specimens or if they relied on the photographs. If so, the authors might explain why looking at the photographs from only two species was enough for their work. Also, what about the other Mexican species compared to A. vega (did the author rely only on the literature?)

In the Discussion, the authors emphasize the asymmetry of the specimen in the palps and legs but do not discuss this issue much further. Given that the authors chose this angle of highlight, it should be more discussed.

6. A large part of the description is composed of measurements (antennomeres, tarsomeres, which could be summarized in Tables rather than mentioned in the text as it is because it is not easy to read.

Minor point:
Blattaria is used in the title but Blattodea is used in the Introduction. Please homogenize.

Reviewer 6 ·

Basic reporting

The manuscript is based on the description of a fossil cockroach found in amber. The finding is interesting because the genus Anaplecta is poorly known while representing a worldwide distributed group in cockroaches.

The introduction should be improved to better present the study. Making a long list though incomplete of fossil cockroaches studied until today is not useful. The authors should better focus on more recent phylogenetic studies (Legendre et al., 2015; Evangelista et al., 2019) to replace their finding into a wider scope.

The qualifier cosmopolitan for the genus Anaplecta is misleading. The word 'Cosmopolitan' lets the reader thinking that this is a tramp genus present everywhere without local diversification. They should rather suppress it from the title and explains that the genus has a worldwide distribution.

The English could be slightly improved in order to make the manuscript more attractive

Experimental design

The placement of the fossil specimen in the genus is not made according to the relevant literature. Grandcolas (1996) has been providing several apomorphies for the family Anaplectidae that could be esasily used here in this respect. This author also described wing folding that the present authors are citing as an informative character.

The authors should abandon gradistic characterizations such as primitive or evolved for taxa. Whatever the position more or less deeply nested of Anaplecta within published trees, this does not allow arguing that Anaplecta would be especially primitive or not. This is basic phylogenetic reasoning for which explanations can be found in recent papers (Crisp, M. D., Cook, L. G. 2005. Do early branching lineages signify ancestral traits? Trends in Ecology & Evolution 20, 122-128)

They should refer once again to more recent phylogenetic studies (e.g., Legendre et al., 2015) that also document the position of Anaplectidae in cockroaches with a much more relevant sampling that the older study they referred to.

Validity of the findings

The finding seems correct but the characters used should be attributed to previous publishing authorities.

Additional comments

The paper is trying to make it worth publishing of a new species of fossil cockroach belonging to a poorly known genus. Though useful as showing an original finding, the paper should be freed for the problems evoked above. The authors should cite all relevant literature. Within the discussion, they could usefully make a review of the knowledge - recent and less recent - existing on the genus Anaplecta.

---

## Round 0.2 · Major Revisions

The one reviewer (R6) suggested the paper needs a major revision. If the authors can focus on not only the new species reported but also the evolution of the new species, the paper will be more values.
Please consider this reviewer's suggestion.

Reviewer 3 has provided an annotated PDF for you

Reviewer 3 ·

Basic reporting

no comment

Experimental design

no comment

Validity of the findings

no comment

Additional comments

I provide a pdf where I tried to address and/or correct some errors in the text.

Annotated reviews are not available for download in order to protect the identity of reviewers who chose to remain anonymous.

Reviewer 6 ·

Basic reporting

The manuscript has been improved since the first reviewing session. However, it still remain some aspects that should be corrected to make it acceptable for publication.
The introduction is still a very biased catalogue of the works of some authors while some others, largely influential (e.g., Legendre et al., 2015), are still not cited.

The authors have corrected many details but they still failed to give the manuscript a very general value, further than the description of one peculiar fossil species of Anaplectidae. At this stage, this is up to the editor to decide whether PeerJ should publish it or defer it to a journal of taxonomy accepting pieces of one-species taxonomy.

Experimental design

This is OK

Validity of the findings

Ok again

Additional comments

The authors have corrected many details but I feel that they still failed to give the manuscript a very general value, further than the description of one peculiar fossil species of Anaplectidae.

---

## Round 0.3 · Minor Revisions

Your revision is now scientifically Acceptable, however the Section Editors feel that the paper needs considerable editing for English grammar prior to publication. Therefore, please revise the paper one final time for English language issues.

---

## Round 0.4 · accepted · Accept

The manuscript has been accepted for publication. Congratulations!